# Night-Time Shift Work and Related Stress Responses: A Study on Security Guards

**DOI:** 10.3390/ijerph17020562

**Published:** 2020-01-15

**Authors:** Emanuele Cannizzaro, Luigi Cirrincione, Walter Mazzucco, Alessandro Scorciapino, Cesare Catalano, Tiziana Ramaci, Caterina Ledda, Fulvio Plescia

**Affiliations:** 1Department of Sciences for Health Promotion and Mother and Child Care “Giuseppe D’Alessandro”, University of Palermo, via del Vespro 133, 90127 Palermo, Italy; luigicirrincione@gmail.com (L.C.); walter.mazzucco@unipa.it (W.M.); fulvio.plescia@unipa.it (F.P.); 2Department of Prevention, Area of Protection and Safety in the Workplace, Provincial Health Authority, 95124 Catania, Italy; alescorc@gmail.com (A.S.); catalanocesare@gmail.com (C.C.); 3Faculty of Human and Social Sciences, Kore University of Enna, 94100 Enna, Italy; tiziana.ramaci@unikore.it; 4Clinical Pathology, ARNAS “Garibaldi”, 95123 Catania, Italy; cledda@unict.it

**Keywords:** occupational stress, HPA axis activation, work-related stress, anxiety-like behavior

## Abstract

Work-related stress can induce a break in homeostasis by placing demands on the body that are met by the activation of two different systems, the hypothalamic–pituitary–adrenal axis and the sympathetic nervous system. Night-shift work alters the body’s exposure to the natural light–dark schedule and disrupts circadian (daily) rhythms. The greatest effect of night-shift work is the disruption of circadian rhythms. The impact that these disruptions may have on the pathogenesis of many diseases, including cancer, is unknown. This study aims to discover the relationship among three different job activities of security guards and their stress-related responses by evaluating salivary cortisol levels and blood pressure. Methods: Ninety security guards, including night-time workers and night-time and daily-shift workers, were recruited for this study. Each security guard provided two saliva samples before and after three scheduled time points: (i) at 22:00, (ii) at 06:30, and (iii) at 14:00. Results: The results of the study showed a significant alteration in cortisol levels. Night-time shift cortisol levels significantly increased before and after the work shifts. A physiological prevalence of the vagal tone on the cardiocirculatory activity was found during night-shift work. Conclusions: This study indicates that cortisol levels and blood pressure are sensitive markers of biological responses to severe work stress. Shift-change consequences may occur at the end of the night shift when there is a significant increase in the cortisol level and a significant variation in cardiovascular parameters.

## 1. Introduction

Stress is a complex process affected by environmental and psychosocial factors. It initiates a cascade of information processing in both the peripheral nervous system and the central nervous system (CNS) [1]. Stress can be acute (short-lasting) or chronic (repetitive or occurring over an extended period of time) [2]. Under chronic stress conditions, the body remains in a constant state of ‘overdrive’ which causes deleterious effects on the regulation of stress response systems as well as of various of body’s organs [3,4].

Work-related stress is considered a major risk factor for the onset of physical and mental health disorders such as cardiovascular diseases, metabolic syndrome, depression, cognitive impairment and cancer [5,6,7]. Work pressure may be associated with factors like the use of drugs, respiratory tract infections, etc., that may contribute to increase the stress response during the course of human life [8,9,10,11,12,13,14]. Workers define the stress they perceive at work as a “sense of fatigue” [15,16]. Work-related risk factors including high work demands, low level of job control, the role of a worker within an organization, and relationships with co-workers and supervisors should be evaluated to improve the assessment of work-related stress [17].

The physiological stress response is one of the probable mediators of the effects of psychosocial factors on cancer progression. The stress response involves the activation of several body systems including the autonomic nervous system and the hypothalamic–pituitary–adrenal (HPA) axis. 

The fight-or-flight response is elicited by the production of mediators such as norepinephrine (NE) and epinephrine (E) from the sympathetic nervous system (SNS) and the adrenal medulla. The HPA response includes the secretion of corticotropin-releasing hormone from the hypothalamus, which induces the secretion of adrenocorticotrophic hormone from the anterior pituitary, resulting in the production of glucocorticoids (GCs) such as cortisol from the adrenal cortex [18]. Additional neuroendocrine factors are also modulated by stress, including dopamine (DA), prolactin, nerve growth factor (NGF), substance P, and oxytocin [19,20,21,22,23,24,25,26,27]. The HPA axis also plays a pivotal role in homeostatic processes. Furthermore, it is supposed that a dysregulation may be involved in the pathogenesis of cardiovascular disease, type 2 diabetes, and stroke [28,29].

Stress exposure also provokes a shift in many neurobehavioral processes, such as anxiety/vigilance, memory, reward salience, pain sensitivity, and coping behavior. These changes in biological functions produce coordinated and highly adaptive responses that are ascribable to survival in response to a threat [30,31,32]. 

The fast-growing productivity of modern society demands work across time zones. Night-shift work is increasingly prevalent in variousindustries such as food production, entertainment, security, health care, and transportation [33,34,35]. Circadian rhythm disruption induced by electric lighting during night shifts poses huge challenges for public health, increasing the risk of cardiovascular diseases, neuropsychiatric and endocrine system disorders, and even cancers, in particular breast cancer [36,37,38]. Cancer incidence in industrialized areas is considerably higher than in developing countries, suggesting that environmental factors of the modern society play a role in cancer etiology. 

Since 2007, the International Agency for Research on Cancer (IARC) has identified “shift work involving circadian rhythm disruption” to be probably carcinogenic (Group 2A) on the basis of “limited evidence on humans for the carcinogenicity of shift work involving night work” and “sufficient evidence on experimental animals for the carcinogenicity of light during the daily dark period (biological night)” [37,39].

Glucocorticoids are essential for the regulation of immune and inflammatory responses. Physiological concentrations of GCs in the range of 350–950 nmol/L, occurring during physical or psychological stress, result in the modulation of the transcription of genes involved in inflammatory response. The activation of the HPA axis and the release of cortisol are considered significant components of physiological stress in humans [30,31]. 

Salivary cortisol level has proven to be a valid and reliable reflection of the level of unbound hormone in blood. The assessment of cortisol in saliva is widely accepted and frequently used in psychoneuroendocrinology. Due to several advantages, saliva cortisol assessment can be chosen as a biomarker of psychosocial stress and related mental or psychological disorders, reflecting the activity of the HPA axis. Plasma-free cortisol concentration proficiently indicates a robust circadian rhythm, with peak levels in the morning after waking followed by decreasing levels throughout the day. Saliva collection is non-invasive, and salivary biomarkers have the further advantage of being suitable for self-collection [40,41,42,43,44,45,46,47,48].

Based on these discoveries, this study aimed to find the relationship among three different work activities carried out by security guards, the effect of shift work, and stress-related responses by evaluating salivary cortisol levels. We also analyzed the correlation between HPA axis responses and comorbidity with cardiovascular diseases. We hypothesized that stress increased during the work shift, especially in workers involved in night-time operations.

## 2. Materials and Methods 

This research was part of a project evaluating the relationship among different work activities and related stress responses. In total, 240 (100%) male security guards, belonging to a large Sicilian security company, were invited to participate in the study by their management. Since 30% (n = 72) of them refused to participate, the study initially recruited 168 (70%) workers. All participants were informed about the aim of the study and signed informed consent forms before taking part in it. Respondents were asked not to mention their name or the name of their organization in the questionnaire to ensure privacy and anonymity. Data were collected in April 2018, when the temperature was 18 °C to 24 °C.

All participants underwent medical examination before the study began to collect clinical and anamnestic information. This included information about age, occupational history, medical history, physiological anamnesis, current and past medical situation.

At the end of the first medical evaluation, we selected 90 (37.5%) workers (age ranging from 35 to 50 years) without cardiovascular disease and who had not consumed drugs for at least 15 days before the study began and divided them in 2 groups based on their tasks: night-time workers employed in monitoring rooms or inspecting the streets and daily operative workers.

The workers were then further divided in the following groups: 30 night-time workers (22:00–06:00) employed in monitoring rooms (NWM); 30 night-time operative workers (22:00–06:00) inspecting the streets (NWO), and 30 daily operative workers (06:00–22:00) randomly selected as control group (DW). All data were managed according to the Italian law for privacy protection (Decree n. 196, January 2003). The authors obtained authorization by our Ethical Committee. A multidisciplinary team of health experts collected and analyzed the data using several instruments. On the first visit, all workers were questioned about their medical history and underwent a physical examination. Pressure measurements and electrocardiogram (EKG) were acquired by instrumental evaluation from all patients.

### 2.1. Saliva Samples Collection, Cortisol Determination, and Systolic/Diastolic Evaluation

Saliva cortisol was collected three times (Figure 1). Cortisol levels were determined from saliva samples, representing the unbound biologically active hormone fraction. Salivary cortisol levels are highly correlated with total serume-free cortisol levels and are independent of saliva flow rate. A salivette sampling device was used for ease and hygiene reasons. Participants chewed a salivette for 1 min for each measurement. All samples were stored at −20 °C until being assessed. Saliva samples were brought to a laboratory and processed to determine the cortisol levels. Free-cortisol levels were measured using a commercial cortisol saliva Elisa (LDN, Nordhorn, Germany). At the same time of cortisol collections, the diastolic/systolic blood pressure was evaluated (Figure 1).

### 2.2. Statistical Analysis

Data obtained from salivary cortisol and systolic/diastolic pressure determination were analyzed by a one-way ANOVA, followed by Tukey’s multiple-comparison post-hoc test. Data are reported as mean ± S.D. Statistical significance was set at *p* < 0.05. Statistical analysis was performed using GraphPad Prism statistical system (version 7).

## 3. Results

### 3.1. Cortisol Levels

The potential relationship between perturbation of the circadian rhythm and type of work done by the security guards was assessed through salivary cortisol evaluations. The results of a one-way ANOVA performed on salivary cortisol levels showed a significant effect of the different job types at different times: F_(2,87)_ = 6.490; *p* < 0.0024 at Time 0, F_(2,87)_ = 22.37; *p* < 0.0001 at Time 1. No statistical differences were recorded at Time 2 (F_(2,87)_ = 0.8466; *p* = 0.4324).

In detail, Tukey’s multiple-comparison post-hoc test showed a significant increase in salivary cortisol levels at Time 0 in MWM (*q* = 4.067, *p* < 0.05) and in NWO (*q* = 4.691, *p* < 0.01) compared to DW (Figure 2A). At Time 1, the statistical analysis indicated a higher value of salivary cortisol in NWM and MWO (*q* = 6.496, *p* < 0.001; *q* = 9.203, *p* < 0.001) compared to DW (Figure 2B).

No statistical differences were detected in salivary cortisol evaluations in NWM compared to NWO and DW (*q* = 1.364, *p* > 0.05; *q* = 1.752, *p* > 0.05) and in NWO (*q* = 0.3882, *p* > 0.05) compared to DW at Time 2 (Figure 2C).

### 3.2. Systolic and Diastolic Blood Pressure

Systolic and diastolic blood pressure was recorded during the three different work shift times. The results of one-way ANOVA showed a significant variation in systolic and diastolic pressure at Time 1 (F_(2,87)_ = 36.66; *p* < 0.0001; F_(2,87)_ = 25.29; *p* < 0.0001) and Time 2 (F_(2,87)_ = 1.432; *p* < 0.0001; F_(2,87)_ = 1.411; *p* < 0.0001). Tukey’s multiple-comparison test showed a difference in systolic and diastolic pressure in NWM and NMW only at Time 1 (see Table 1).

## 4. Discussion

The current study was undertaken to evaluate the consequences of three different work activities performed by security guards on the circadian rhythm of cortisol and systolic/diastolic blood pressure. 

It is known that, under physiological conditions, cortisol is released according to a circadian rhythm characterized by the lowest level at the night, the maximum level in the early morning in response to events such as waking up, and a progressive decrease reaching the minimum value by the end of the day [49,50].

The results obtained showed that the circadian rhythmicity of cortisol increase in security guards is influenced by night-time shift work and by the different activities performed. This confirms previous research indicating that the endogenous physiological feedback of cortisol is influenced by both the working conditions that alter the sleep waking rhythm and the risk of the activity performed [45].

Our data showed significant quantitative changes in cortisol levels, especially in night-shift workers who inspect the streets, whose cortisol levels tended to increase significantly both before and after the shift.

The factors involved in these responses may be different. In addition to sleep deprivation, it is believed that the particular psychophysical commitment required by the activity conducted on the road plays a fundamental role [49,50,51,52,53,54]. Previous research indicated that exposure to shift work during one’s occupational career causes disruption of the circadian time. by physiologically interacting with the circadian clock. This may result in impaired HPA axis functionality, which regulates the biological response to stressful stimuli [55] such as those that security guard night workers are subjected to. 

Furthermore, the values of the systolic/diastolic blood pressure were reduced in the morning in night-time shift workers, while they did not show changes in the afternoon and evening, compared to those of daily workers. These results suggest that, although the cardiocirculatory activity is substantially controlled by an endogenous circadian rhythm, the inversion of the sleep/wake rhythm, added to a psychophysically stressful working condition, accentuates the physiological prevalence of the vagal tone at night at the end of the shift [56].

The results of this study indicated how cortisol, in night-time shift workers exposed to stressful situations, could be considered a sensitive indicator of biological responses to high work-related stress and to the effects induced by activities altering the normal sleep–wake rhythm. The detection of salivary cortisol proved to be reliable and preferable to the evaluation of blood cortisol, due to its low invasiveness and to the high compliance of the analyzed subjects [41,42,43,44,45,46,47,48].

Furthermore, the use of this simple salivary test could be useful to monitor cortisol levels as an early marker in the prevention or assessment of cancer growth and metastasis. This is important considering recent studies that have identified night-shift work as “probably carcinogenic to humans” (group 2A) [57], and have suggested that chronic stress provokes the activation of specific tumor-promoting signaling pathways in cancer cells and the tumor microenvironment [4].

There are some limitations in this study which should be considered in future research. For example, due to the limited sample size, generalizing the relationship between night-time shift work and the related cardiovascular stress responses is difficult. Understanding the correlation between job-related stress and cardiovascular biomarkers could contribute to implementing intervention and prevention strategies aimed at improving the working conditions and the health of individual workers.

## Figures and Tables

**Figure 1 ijerph-17-00562-f001:**
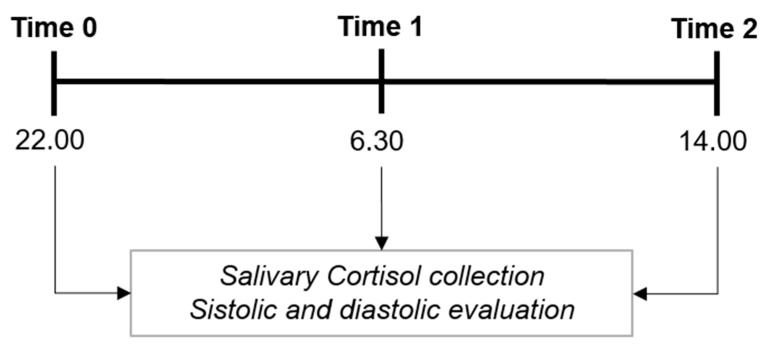
Timeline showing the three times when saliva samples were collected and systolic/diastolic pressure was evaluated. The first evaluation time was approximatively at 22:00 (Time 0), the second one at 06:30 (Time 1), and the third one at 14:00 (Time 2).

**Figure 2 ijerph-17-00562-f002:**
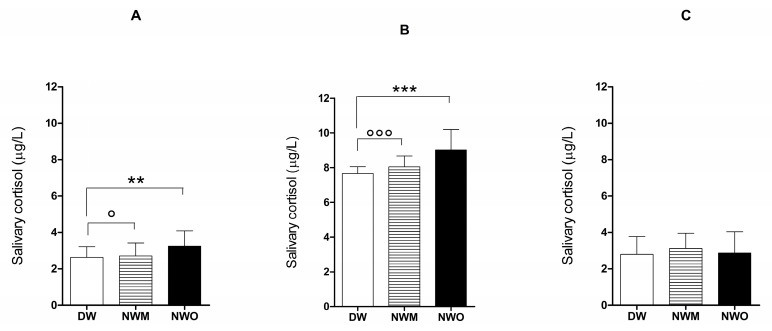
Correlation between different job types and related salivary cortisol levels. Data from a post-hoc Tukey’s multiple-comparison test performed on the effects of shift work on salivary cortisol levels at time 0 (**A**), time 1 (**B**), and time 2 (**C**). Each value represents the mean ± S.D. of 30 subjects. DW: daily workers, NWM: night-time monitoring workers, NWO: night-time operative workers. *** *p* < 0.001, ** *p* < 0.01 vs. DW; °°° *p* < 0.001, ° *p* < 0.05 vs. DW.

**Table 1 ijerph-17-00562-t001:** Results of Tukey’s multiple-comparison test performed on systolic and diastolic pressure at the three different time points. Each value represents the means ± S.D. of 30 subjects.

Systolic Blood Pressure				Diastolic Blood Pressure			
**Time 0**				**Time 0**			
	*value*	*q*	*p-value*		*value*	*q*	*p-value*
***DW***	119 ± 9.113	1.743	*p* > 0.05 vs. NWM	***DW***	83 ± 4.115	1.395	*p* > 0.05 vs. NWM
***NWM***	116 ± 12.77	0.3152	*p* > 0.05 vs. NWO	***NWM***	81 ± 6.084	2.596	*p* > 0.05 vs. NWO
***NWO***	120 ± 13.39	1.427	*p* > 0.05 vs. NWO	***NWO***	80 ± 7.25	1.201	*p* > 0.05 vs. NWO
Time 1				Time 1			
	*value*	*q*	*p-value*		*value*	*q*	*p-value*
***DW***	114 ± 8.715	8.642	*p* < 0.001 vs. NWM	***DW***	74 ± 5.792	8.085	*p* < 0.001 vs. NWM
***NWM***	102 ± 7.90	11.67	*p* < 0.001 vs. NWO	***NWM***	66 ± 5.470	9.224	*p* < 0.001 vs. NWO
***NWO***	97 ± 7.10	3.026	*p* > 0.05 vs. NWO	***NWO***	64 ± 6.584	1.139	*p* > 0.05 vs. NWO
Time 2				Time 2			
	*value*	*q*	*p-value*		*value*	*q*	*p-value*
***DW***	120 ± 5.107	1.846	*p* > 0.05 vs. NWM	***DW***	80 ± 15.26	2.16	*p* > 0.05 vs. NWM
***NWM***	117 ± 13.08	2.242	*p* > 0.05 vs. NWO	***NWM***	75 ± 10.81	0.2244	*p* > 0.05 vs. NWO
***NWO***	118 ± 5.949	0.3954	*p* > 0.05 vs. NWO	***NWO***	79 ± 9.01	1.936	*p* > 0.05 vs. NWO

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
