# Peer review of "Night-Time Shift Work and Related Stress Responses: A Study on Security Guards"

_ijerph, 2020, doi:10.3390/ijerph17020562_

Round 1
Reviewer 1 Report
The authors presented the purpose of the own research as follows:
The present study aimed to investigate the relationship between three different work activities carried out by the security guards, the effect of shift works and related stress responses, by salivary cortisol evaluations. We also analyzed the correlation between HPA axis responses and the comorbidity with cardiovascular diseases. We hypothesized that stress increased in the guards and that, this amount, might be directly correlated with workers especially involved in nighttime operations.
In my opinion, the study was poorly planned and none of the goals were achieved.
One cannot conclude the level of stress only on the basis of a very general description of the work performed: "nighttime workers employed in monitoring rooms", "nighttime operatives inspecting the streets", "daily operative workers". If you want to study stress responses, you need to study not only reactions but stress levels, and this has not been done.
There is no analysis in the article of the correlation between HPA axis responses and the comorbidity with cardiovascular diseases.
In addition, attempts to interpret low blood pressure values in security guards working at night are misguided, and in any case not supported by literature data.
In my opinion, the possibility of publishing this article should not be considered.
Author Response
Title: Nighttime shift work and related stress responses: study on security guards
Ref: ijerph-671182
REW. TABLE NUMBER 1
|
Number of comment |
Comment of first peer-reviewer |
Summary of responses to Editor and the peer reviewers' comments |
|
|
||
|
Extensive literature |
In my opinion, the study was poorly planned and none of the goals were achieved. 1) One cannot conclude the level of stress only on the basis of a very general description of the work performed: "nighttime workers employed in monitoring rooms", "nighttime operatives inspecting the streets", "daily operative workers". If you want to study stress responses, you need to study not only reactions but stress levels, and this has not been done.
|
Thank you and the Reviewer for your feedback and constructive criticism on the above manuscript, which has modified based on the suggestions received. We hope this version may be viewed favorably. We clarify and better try to explain this concept: “Salivary cortisol has proven to be a valid and reliable reflection of the respective unbound hormone in blood. Assessment of cortisol in saliva is a method widely accepted and frequently used in psychoneuroendocrinology. Due to several advantages saliva cortisol assessment can be chosen as biomarker of psychosocial stress and related mental or psychological disorder, demonstrating activity of the HPA axis” “The detection of salivary cortisol therefore proved to be reliable and preferable, compared to the evaluation of blood cortisol, due to its low invasiveness and the high compliance of the analyzed subject” We know that if we want to study stress responses we need to study not only reactions but stress levels. Anyway we preferred this approach linked to our research interests. To better explain our point of view, and in order to respond your good consideration we underline that, as many researches indicate, occupational stress can also be assessed by salivary cortisol levels. We add the references as above: • Šušoliaková et al. 2018. Assessment of work-related stress by using salivary cortisol level examination among early morning shift workers. Cent Eur J Public Health. 26(2):92-97. • Weibel et al., 1996. Internal dissociation of the circadian markers of the cortisol rhythm in night workers. Am J Physiol • Marchand et al., 2015. Work stress models and diurnal cortisol variations: The SALVEO study. J Occup Health Psychol. ;21(2):182-93. • Kirschbarum et al.,1994. Salivary cortisol in psychoneuroendocrine research: Recent developments and applications. Psychoneuroendocrine. 19: 313-333.
|
|
|
2)There is no analysis in the article of the correlation between HPA axis responses and the comorbidity with cardiovascular diseases.
|
In order to better explain and check your comments we add some analysis:
HPA axis plays a pivotal role in the homeostatic processes. Furthermore, a dysregulation on its activity is also thought to be involved in the pathogenesis of cardiovascular disease, type 2 diabetes and stroke.
|
|
|
3) In addition, attempts to interpret low blood pressure values in security guards working at night are misguided, and in any case not supported by literature data.
|
At this point we, as above, refer to other references uploaded to clarify as our hypothesis is supported by literatures: •Šušoliaková et al. 2018. Assessment of work-related stress by using salivary cortisol level examination among early morning shift workers. Cent Eur J Public Health. 26(2):92-97.
|
|
English revision |
|
Done. |

Reviewer 2 Report
Thank you for the opportunity to review the manuscript “Nighttime shift work and related stress responses: study on security guards” for International Journal of Environmental Research and Public Health. This manuscript reports changes in cortisol levels among security grads working the over night shift. This was an interesting manuscript, and I enjoyed reviewing it. There is much to like with this paper. The experimental design and resulting data are unique and suitable to explore an important topic. Overall, it was thought provoking and an enjoyable read. I do, however, some comments on how the manuscript needs to be improved.
The paper is in need of basic editing, as its present form makes for a very distressing read and at times interferes mightily with comprehension.
The authors need to clarify the details regarding the recruitment process. For example, what is described in the manuscript, to my reading, could best be described as a convenience sample. As such, these data were drawn using a nonprobability sampling design and, consequently, cannot be generalized beyond the sample. There is nothing inherently wrong with this approach, especially considering this is an exploratory study (which, by the way, should be stated in the methods section). In my view the scholarship of this work would be strengthened considerably if the author(s) would be more explicit explaining sampling decisions.
Related to my previous point, without knowing the specific details regarding the recruitment process, I am unable to accurately interpret the reported findings. Please update in light of modifications tied to point 2 above.
Author Response
Title: Nighttime shift work and related stress responses: study on security guards
Ref: ijerph-671182
REW. TABLE NUMBER 2
|
Number of comment |
Comment of first peer-reviewer |
Summary of responses to Editor and the peer reviewers' comments |
|
|
||
|
|
1) The paper is in need of basic editing, as its present form makes for a very distressing read and at times interferes mightily with comprehension.
|
We are grateful to the Reviewer for positive and encouraging feedback. The issues raised are addressed here below in a point-by-point fashion; amendments made to the manuscript are in red color.
Basic editing has been done |
|
|
2) The authors need to clarify the details regarding the recruitment process. For example, what is described in the manuscript, to my reading, could best be described as a convenience sample. As such, these data were drawn using a nonprobability sampling design and, consequently, cannot be generalized beyond the sample. There is nothing inherently wrong with this approach, especially considering this is an exploratory study (which, by the way, should be stated in the methods section). In my view the scholarship of this work would be strengthened considerably if the author(s) would be more explicit explaining sampling decisions. |
We specified specific details regarding the recruitment process, and updated in light of modifications done.
Effectively we have selected 90 participants in a population of 243 workers based on their state of health. Following the reviewer’s suggestion, we specified the recruitment process in the materials and methods.
|
|
English revision |
|
Done. |
I look forward to Your feedback.
Best regards
Palermo, 02 January 2020

Round 2
Reviewer 1 Report
I accept the explanations and modifications made by the authors. I believe that the manuscript has been significantly improved and can now be published in IJERPH.
Author Response
As suggested we have corrected English language and style
Reviewer 2 Report
Thank you for the opportunity to review the revised manuscript “Nighttime shift work and related stress responses: study on security guards” for International Journal of Environmental Research and Public Health. For the most part, the authors, in professional manner, addresses the concerns I raised in my original review. However, there was little effort to update the discussion section to reflect the revisions made in explaining sampling decisions. As such, the discussion section is rife with misleading and factually incorrect statements. The described in the manuscript were drawn using a nonprobability sampling design and, consequently, cannot be generalized beyond the sample. To wit, the results of the study are not generalizable and, therefore, speak exclusively to the data under investigation. Consequently, as currently written, the authors’ conclusions are not supported by the data/methods presented. This is not a small issue. This must be addressed.
Author Response
As suggested we have corrected English language and style
Thank you for your feedback and constructive criticism on the manuscript “Nighttime shift work and related stress responses: study on security guards”, which has modified based on the suggestions received.
Round 3
Reviewer 2 Report
The authors have addressed my concerns. Thank you.